# CVeDRL: An Efficient Code Verifier via Difficulty-aware Reinforcement Learning

## Abstract

Code verifier is the key to the post-verification process in large language model (LLM) code generation. However, supervised fine-tuning (SFT) methods suffer from dataset scarcity, high error and failure rates, and severe inference delay. In this work, we adapt reinforcement learning to train an efficient code verifier, CVeDRL, which substantially alleviates these challenges and balances performance and efficiency in only a 0.6B scale. First, we design syntax and functionality rewards and employ GRPO to train the base code verifier. However, preliminary experiments indicated that the base model could not produce effective unit tests for difficult branches and samples. Then we propose Branch-Difficulty-aware and Sample-Difficulty-aware reinforcement learning based on exponential reward shaping and static analysis metrics (Halstead Complexity and Maintainability Index). Experimental results show that CVeDRL significantly outperforms the vanilla model while remaining competitive with state-of-the-art models such as GPT-4o-mini and GPT-3.5 in pass rate, assertion failure rate, and code coverage, etc. Furthermore, CVeDRL-0.6B improves inference efficiency by more than 20x compared with LLM trained with SFT method. Code is available at https://anonymous.4open.science/r/CVeDRL-DF1A/

## 1 Introduction

Recently, large language models (LLMs) have shown impressive capabilities in code generation (Achiam et al., 2023; Dubey et al., 2024; Jaech et al., 2024; Guo et al., 2025). Although LLMs are able to quickly generate code solutions, they struggle to produce correct code in a single attempt. Researchers investigate inference-time scaling methods to alleviate this difficulty, where LLMs first employ repeated sampling to output multiple code results, and then a code verifier selects the final best result (Lightman et al., 2024b; Brown et al., 2024). Consequently, the performance of the code verifier is the key to the success of the code generation task for large language models (Cobbe et al., 2021; Lightman et al., 2024b; Liu et al., 2025; Zhao et al., 2025).

Code verifiers take the problem description and the candidate code solution as input and leverage LLMs to generate unit tests, which consist of reasonable input and corresponding output pairs. By executing candidate code solutions and their unit tests, the execution outcomes are examined from compilers and interpreters to identify the optimal code solutions among all candidates. In fact, code verifiers are specific LLMs for code generation, as unit tests are special code snippets. As a result, instruction supervised fine-tuning (SFT) can endow LLMs with unit test generation ability and make them qualified code verifiers (Ma et al., 2025). However, there are three challenges in the SFT strategy. First, large-scale high-quality SFT data is unavailable for unit test generation. Although Ma et al. (2025) presents an automatic data pipeline, a lot of incorrect unit tests exist in the final results. Second, the SFT-based model suffers from a high error rate and failure rate on generated unit tests. Besides, SFT-based code verifiers have to repeatedly sample multiple unit tests to mitigate error and failure issues, causing a serious efficiency bottleneck.

The reinforcement learning (RL) training paradigm induces and supervises LLMs to explore appropriate answers through delicate reward signals, which shows substantial potential in the LLM post-training period (Jaech et al., 2024; Shao et al., 2024; Guo et al., 2025). Training efficient code verifiers by RL is a promising approach to address the above challenges. For one thing, RL training only needs the problem description and the candidate code, without relying on corresponding unit

tests, which avoids the requirement of high-quality datasets. For another thing, common metrics such as pass rate and line coverage of unit tests in software engineering can be adapted as RL reward functions, directly reducing error and failure rates during training. In addition, unleashing the post-training capability of small-scale LLMs and decreasing repeated sampling by RL contribute to efficiency improvement.

In this work, we adapt reinforcement learning methods to train an efficient code verifier CVeDRL, which has both performance and efficiency advantages at a scale of 0.6B. To ensure that the generated unit test cases align with the formatting requirements of test suites and achieve extensive branch coverage for enhanced test quality, we propose a syntax-functionality composite reward and utilize Group Reward Policy Optimization (GRPO) Shao et al. (2024) for training. Specifically, we theoretically analyze the relationship among test-case pass rates, branch coverage, and the efficacy of the code verifier. However, preliminary experiments indicated that our initial approach inadequately distinguished boundary branches and variations in sample difficulty associated with code solutions. Consequently, we introduce a branch-difficulty-aware mechanism grounded in exponential reward shaping and sample-difficulty-aware technique with static analysis metrics to handle both problems respectively.

We perform extensive experiments to verify CVeDRL across three datasets and four policy models. Experimental results demonstrate that CVeDRL-0.6B significantly improves the performance of open-source policy models and also enhances the effectiveness of closed-source models. Additionally, to assess the intrinsic quality of generated unit tests, we directly evaluate various metrics such as error rates, pass rates, and line coverage across three datasets. CVeDRL-0.6B consistently outperforms various baseline models in the task of unit test generation, achieving a considerably higher test-case pass rate on MBPP+ (yielding a 17.55% increase compared to GPT-4o-mini) with marginally higher coverage, and substantiating our theoretical analyses. Furthermore, CVeDRL-0.6B gains more than 20x inference efficiency improvement on token throughput, compared to the traditional SFT model CodeRM.

The contributions of our work are as follows:

- We theoretically analyze the interplay among test case pass rates, branch coverage, and the performance of code verifiers.
- We propose CVeDRL, an efficient code verifier trained by GRPO with syntax and functionality rewards, alongside a novel branch-sample difficulty-aware mechanism incorporating exponential reward shaping and static analysis metrics.
- Extensive experiments demonstrate that CVeDRL-0.6B achieves sota performances on LLM code verification and unit test generation across six groups of main experiment overall and improves inference efficiency by more than 20x compared with SFT models.

## 2 RELATED WORK

**Enhancement of code generation via unit testing.** Enhancing the reliability of code generation with unit test cases can be framed as a two-phase process: improving model training and refining inference. Extensive prior work integrates unit test results into the training process to improve the accuracy of code generators Liu et al. (2023a); Dou et al. (2024). CURE Wang et al. (2025c) demonstrates that co-training tests and code yields more robust behavior in the training phase, and subsequent RL variants incorporate real execution feedback from unit test runs to further shape model policies. Prior research validates codes generated by LLM with automated testing at inference time with optimal solution selection. For instance, MBR-EXEC Shi et al. (2022b) applies Bayesian risk decoding to rerank code candidates by their pass rates. CodeRM Ma et al. (2025) leverages a distilled "test generator" model to produce targeted unit tests that guide solution selection. Building on these insights, our method combines test accuracy and line-coverage signals within a unified reward function, thereby both sharpening higher quality of unit test generation and enabling more reliable optimal code selection at inference.

**Reinforcement Learning for LLMs.** Recent work systematically frames text generation as a Markov decision process, adapting the classical RL pipeline to large language models by treating each token prediction as an action and defining rewards that capture response quality Wang et al.

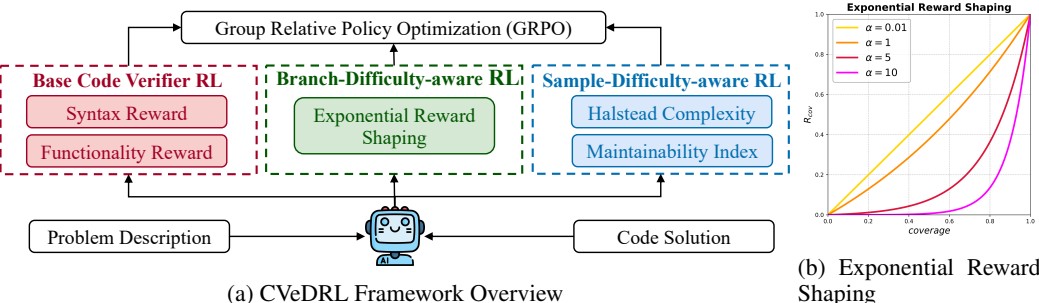

(a) CVeDRL Framework Overview

(b) Exponential Reward Shaping

Figure 1: (a) We design syntax and functionality rewards and employ GRPO to train the base code verifier. Base model struggles to produce effective unit test cases for the difficult branches and samples. Therefore, we propose Branch-Difficulty-aware and Sample-Difficulty-aware reinforcement learning based on exponential reward shaping and static analysis metrics (Halstead Complexity and Maintainability Index). (b) Exponential reward shaping modifies the coverage reward function from a linear format into an exponential format.

(2025a). Prominent architectures extend base models (e.g., Alpaca, LLaMA) via policy-gradient, proximal policy optimization, and actor-critic algorithmsDong et al. (2025) to iteratively enhance coherence and factuality Hu et al. (2025). Two main reward-model techniques, Reinforcement Learning from Human Feedback (RLHF) Christiano et al. (2023) using curated human preferences and Reinforcement Learning from AI Feedback (RLAIF)Lee et al. (2024), have become standard for aligning outputs with desired behaviors. Direct Preference Optimization (DPO) Rafailov et al. (2024) methods bypass explicit reward functions by directly fitting model parameters to preference data, achieving comparable alignment with lower complexity. Recent extensions of DPO include Group Robust Preference Optimization (GRPO) Ramesh et al. (2024), which adaptively re-weights group-specific losses to ensure worst-case group performance under data imbalance. Balanced Preference Optimization (BPO) Wang et al. (2025b) applies dynamic reward-margin balancing to stabilize preference updates in DPO.

**Unit test generation.** Unit test generation automates the creation of test cases to verify code correctness, expedite bug detection, and uphold source quality. Traditional approaches often employ symbolic analysis Galeotti et al. (2013) and meta-heuristic Harman & Jones (2001); McMinn (2004); Harman et al. (2012) algorithms to craft tests. LLM-based techniques have garnered attention for their efficiency, interpretability, and readable outputs Jiang et al. (2024). Prior research treat LLM as an auxiliary part for traditional methods to help the exploration Lemieux et al. (2023) or mutation Brownlee et al. (2023). More recently, LLM-based methods fall into two main categories: prompt engineering and model fine-tuning. Prompt engineering frameworks include decomposing test objectives into sub-questions Wang et al. (2024), offering extra static metrics to LLM in prompt Sepidband et al. (2025), and iteratively incorporating execution feedback into prompt to emulate human debugging Pizzorno & Berger (2025); Chen et al. (2024); Cheng et al. (2025). Alternatively, LLMs can be fine-tuned for the specific task of test generation Ma et al. (2025); Eom et al. (2024). In this work, we propose a reinforcement learning–based strategy that robustly integrates dynamic execution feedback with static code analysis to produce high-quality unit tests.

## 3 METHOD

In this section, we present CVeDRL, which has both performance and efficiency advantages at a scale of 0.6B. We begin by introducing the Unit Test Majority-Voting Framework and its associated reliability bound of test case quality. Then we introduce the base code verifier trained by syntax and functionality rewards for GRPO. To handle boundary conditions, we apply an exponential reward shaping mechanism that amplifies rewards for covering rare branches. Additionally, we integrate two static analysis metrics that provide priori code sample complexity assessments, refining the reward function with static insights.

## 3.1 Unit Test Majority-Voting Framework and Confidence Bound

The unit test majority-voting framework adopts the well-established best-of-N decoding strategy, wherein an LLM policy model first generates $N$ candidate programs for a given programming problem $Q$ (Cobbe et al., 2021; Lightman et al., 2024a). Formally, we denote these candidates as $\{s_1, s_2, \ldots, s_N\}$. To assess functional correctness, an auxiliary LLM produces $M$ unit tests for each $(Q, s_i)$ pair, yielding test suites $\{T_1, T_2, \ldots, T_M\}$, where each test suite $T_j = \{(x_{j,1}, y_{j,1}), (x_{j,2}, y_{j,2}), \ldots, (x_{j,K_j}, y_{j,K_j})\}$ contains $K_j$ input–output pairs. Here, $x_{j,k}$ is the $k$-th input and $y_{j,k}$ the expected output. Each candidate $C_i$ is executed against all tests $T_j$, producing binary outcomes

$$\mathrm{p}_{i,j} = \begin{cases} 1, & \text{if } s_i \text{ passes every case in } T_j, \\ 0, & \text{otherwise.} \end{cases}$$

These results form a reward vector $\mathbf{p}_i = (\mathrm{p}_{i,1}, \ldots, \mathrm{p}_{i,M})$ for each candidate. Finally, under the majority-voting criterion (Wang et al., 2023), we select the candidate that maximizes the total number of passed tests:

$$\mathrm{s}_{\mathrm{opt}} = \arg \max_{i \in \{1, \ldots, N\}} \sum_{j=1}^{M} \mathrm{p}_{i,j}.$$

To quantify how test-assertion reliability and branch coverage jointly influence the confidence in the selected program, we derive that the assertion correctness probability $p$ must satisfy the bound

$$p \geq \frac{1 + \sqrt{\frac{2}{M} \ln\left(\frac{1-q}{1-q'} N\right)}}{1 + c},$$

where $q'$ denotes the overall probability of correctly selecting a program under the majority-voting framework, $q$ denotes the prior probability that any individual candidate is functionally correct. $N$ indicates the total number of generated programs under consideration, $M$ signifies the number of independent test suites executed for each candidate, while parameter $c$ captures the average branch coverage. By making these dependencies explicit, the bound provides practical guidelines on how to trade off candidate-pool size, test-suite count, and test-generation quality to achieve a target post-selection confidence level $q'$. This result is obtained by applying Hoeffding inequality to the bounded difference of vote counts of correctness, together with a union bound over all candidate comparisons. The detailed proof is in Appendix A.

## 3.2 Base Code Verifier by Reinforcement Learning

Previous studies Xie et al. (2025); Zeng et al. (2025) demonstrate that the design of 'format-answer' reward effectively standardizes the format and ordering of model outputs, which guides the reasoning process of the model. Inspired by these studies, we propose the basic design of reinforcement fine-tuning framework within CVeDRL that dynamic testing feedback via two complementary rewards: Syntax Reward and Functionality Reward. Syntax reward enforces specific AST-derived formatting rules, and Functionality Reward is based on the testing execution results. This approach significantly mitigates the propensity of models to 'hack' evaluations, standardizing generated code format with dynamic test execution signals.

**Syntax Reward.** Inspired by Guo et al. (2025), we reexamine the formatting and syntax requirements for the unit-testing task and designed a corresponding syntax-based reward. Specifically, the generated tests must be strictly enclosed within a final Python code block, and traversing the AST of this test code must reveal at least one class inheriting from `unittest.TestCase`. Given that $W_i$ is the $i^{th}$ concatenation response of correct code solution $s$, the format reward is calculated as follows:

$$r_{syn}(W_i|s) = \begin{cases} 1.0, & \text{if syntax is correct,} \\ -1.0, & \text{if syntax is incorrect.} \end{cases}$$

**Functionality Reward.** Once the syntax is validated, a regex-based extractor retrieves the correctly structured unit-test snippet $u_i$ from the response of model $W_i$. To tailor concepts from prior RL-based fine-tuning methods for reducing hallucination rates specifically, we refine these

approaches by classifying actual testing execution outcomes to derive reward signals. During unit-test generation, models exhibit two primary types of hallucinations: (1) Errors, where wrong code or invalid inputs cause test execution to crash, i.e. both false positives and false negatives; and (2) Failures, where tests run successfully but incorrectly predict the output from source code $s$. Because failures are inherently less severe than errors, we assign distinct negative rewards proportional to each category. For tests is passed, we introduce branch coverage rate for the reward. The execution-result reward is computed as follows:

$$r_{func}(u_i, s) = \begin{cases} -2.0, & \text{if } u_i \text{ is error towards } s \\ -1.5, & \text{if } u_i \text{ is failure towards } s \\ +cov(u_i, s), & \text{if } u_i \text{ is passed} \end{cases}$$

In light of GRPO Shao et al. (2024), for question $q = q(s)$ with corresponding code solution given as the input to LLM, the model samples a group of outputs $\{o_i\}_{i=1}^G$ from the old policy $\pi_{\theta_{old}}$ then optimize the policy with the following objective iteratively:

$$\mathcal{J}(\theta, \{o_i\}_{i=1}^G) = \mathbb{E}_{q \sim P(Q), \{o_i\} \sim \pi_{\theta_{old}}(\cdot|q)} \left[ \frac{1}{G} \sum_{i=1}^G \min\left[ r_i(\theta, \theta_{old}|q)a_{o_i}, \text{clip}(r_i(\theta, \theta_{old}|q), \varepsilon)a_{o_i} \right] \right]$$

$$- \beta \, \mathbb{E}_{q \sim P(Q), \{o_i\} \sim \pi_{\theta_{old}}(\cdot|q)} \left[ D_{KL}(\pi_\theta \| \pi_{ref}) \right],$$

$$r_i(\theta, \theta_{old}|q) = \frac{\pi_\theta(o_i \mid q)}{\pi_{\theta_{old}}(o_i \mid q)},$$

where $\text{clip}(x, \varepsilon) := min(max(x, 1 - \varepsilon), 1 + \varepsilon)$, $\pi_\theta$ is theaa optimal policy. $a_{o_i}$ is the advantage term of $i^{th}$ output calculated by the following formula:

$$a_{o_i} = \frac{r_i - \mu_{r_i}}{\sigma_{r_i}}, r_i = \begin{cases} r_{syn}(W_i|s), & \text{if syntax is incorrect} \\ r_{syn}(W_i|s) + r_{func}(u_i, s), & \text{if syntax is correct} \end{cases}$$

where $\mu_{r_i}$ and $\sigma_{r_i}$ is the mean and standard deviation.

## 3.3 BRANCH-DIFFICULTY-AWARE REINFORCEMENT LEARNING

The Base Code Verifier employs a linear coverage-based reward, which leads the model to favor generating happy-path test cases and to overlook many boundary conditions. We attribute this behavior to the diminishing reward at high coverage levels, resulting in insufficient exploration incentives. To encourage the model to focus more on boundary conditions and other atypical execution paths, we adopt an empirical yet efficient reward design with coverage awareness:

$$r_{cov}(u_i, s) = (e^\alpha - 1)^{-1}[exp(\alpha \times cov(u_i, s)) - 1]$$

where $cov(u_i, s) \in [0, 1]$ is the coverage rate of unit-test case with a given code, and $\alpha > 0$ is a tunable hyperparameter controlling the tail-heaviness of the curve. $r_{cov}$ remains nearly linear for small $\alpha$ but stays flat at low coverage and then rises sharply as $\alpha$ increases. Under this scheme, branches with low coverage (i.e. rare or non-typical paths) receive a much larger incremental reward compared to those already well covered, thereby shifting the focus of policy toward discovering boundary conditions and other special cases after the main path is mastered. By integrating $r_{cov}(u_i, s)$ into our reward signal, we empirically observe that the model devotes additional generation effort to hard-to-reach logic, resulting in more balanced and comprehensive unit-test suites.

## 3.4 SAMPLE-DIFFICULTY-AWARE REINFORCEMENT LEARNING

A key limitation of relying solely on dynamic feedback, such as pass/fail signals or code coverage metrics, is that these measures only become available **after** the model has already generated and attempted to execute a candidate solution. Consequently, they offer no guidance in distinguishing between "trivial failures" (e.g., minor input mismatches or off-by-one errors) and genuinely challenging code fragments that demand deeper reasoning. To address this gap, we introduce a static difficulty prior that quantifies intrinsic complexity of each solution **before** any execution takes place. Specifically, for all code solution, we compute two complementary static metrics of each code during preprocessing: Halstead Complexity($D_H$) and the difficulty of Maintainability($D_M$).

Halstead Complexity, by quantifying the variety and frequency of operators and operands, serves as a metric for the cognitive load required to parse and is widely utilized for code difficulty analysis Hariprasad et al. (2017). The formulation is $\hat{D}_H = \frac{\eta_1}{2} \times \frac{N_2}{\eta_2}$, where $\eta_1$ and $\eta_2$ denote the counts of distinct operators and operands in the source code, and $N_2$ is the total number of operand occurrences. This metric captures the cognitive burden imposed by syntactic diversity and operand repetition. To mitigate the influence of extreme outliers, we collect the set of all $\hat{D}$ values across our training corpus and determine the 95 percentile with a clipped min–max normalization $\hat{D}_{95}$. We then perform a clipped min–max normalization to get Halstead difficulty $D_H$:

$$D_H = min(\hat{D}, \hat{D}_{95})/\hat{D}_{95}.$$

Maintainability Index (MI) Coleman et al. (1994) further captures the anticipated effort and risk of future modifications, which is a four-metric polynomial equation for measuring how maintainable the code is. Recent work Zheng et al. (2024) on MI is applied to the study of code sample differentiation benchmark. The formulation is as follows: $\mathrm{MI} = \max\{0, 100 \times (171 - 5.2\ln V - 0.23G - 16.2\ln L + 50\sin(\sqrt{2.4C}))/171\}$, where $V$ denotes the Halstead Volume, which quantifies the size and information content of the code; $G$ is the Cyclomatic Complexity; $L$ represents lines of code and $C$ is the percentage of comment lines. Since higher $\mathrm{MI}$ values indicate easier maintenance (lower difficulty), we invert and rescale it into a unified difficulty measure of maintainability:

$$D_M = \max\{0,\ 1 - \frac{MI}{100}\} \in [0, 1].$$

Low-maintainability code often deviates from the idiomatic patterns observed during pretraining, thereby undermining the transferability of learned representations and increasing hallucination rates. To emphasize the synergy between code comprehension and maintenance difficulty, we deviate a geometric mean of two metrics $D_H$ and $D_M$:

$$D = \sqrt{D_H \times D_M} \in [0, 1]$$

where indicates the co-occurrence of high values in both dimensions, i.e. only code that is both hard to comprehend and hard to maintain yields a large difficulty.

Motivated by the imperative to distinguish genuinely challenging code fragments from trivial failures prior to execution, we introduce the static difficulty $D$ into our reward. Formally, we define the total augmented execution reward $r_{func}$ with exponential reward shaping in coverage:

$$r_{func}(u_i, s) = \begin{cases} -2.0, & \text{if } u_i \text{ is error towards } s; \\ -1.0 \quad - \ (1 - D), & \text{if } u_i \text{ is failure towards } s; \\ r_{cov}(u_i, s) \ \cdot \ (1 + D), & \text{if } u_i \text{ is passed,} \end{cases}$$

where LLM receives larger positive feedback when the execution is passed with harder code functions, while failures incur softened penalties. This design ensures that the agent allocates a greater exploration effort and learning capacity to high-complexity regions of the code space, thereby improving overall policy robustness on complex unit-testing tasks with static difficulty awareness.

## 4 EXPERIMENTS

### 4.1 EXPERIMENTAL SETUP

Our experiment includes two components: Validation-Coder Performance and CVeDRL Test Quality evaluation. In the evaluation of validation in coder, we employ unit-test reward signals to select among candidate code solutions produced by coder models, which reveals effectiveness of each model on the validation-coder task. In the test quality evaluation, we execute all generated unit tests and report metrics for each large model's output, thereby providing a direct assessment of test-case quality (see in Table 2). We select a compact 0.6B parameter model for CVeDRL training primarily to ensure it can serve as a sufficiently fast and efficient code verifier.

**Datasets.** We utilize four benchmarks to comprehensively evaluate unit-test generation: HU-MANEVAL+ (Liu et al., 2023b), MBPP+ (Liu et al., 2023b), LIVECODEBENCH (Jain et al., 2024),

| Method | Scale | Policy Model | | |
| --- | --- | --- | --- | --- |
| | | Llama3 8B | Llama3 70B | GPT-3.5 GPT-4o-m |
| **HumanEval+** | | | | |
| Vanilla | – | 53.43 | 73.10 | 67.44 82.57 |
| MBR-E | – | 60.18 | 75.47 | 70.53 85.31 |
| CodeT | – | 65.22 | 76.14 | 73.92 85.52 |
| Llama3.1 | 70B | 71.95 | 78.41 | **79.88** 85.48 |
| CodeRM | 8.0B | 72.13 | 78.66 | 78.13 86.49 |
| Qwen3 | 0.6B | 55.75 | 73.91 | 69.16 83.01 |
| Qwen3 | 32B | 70.64 | 78.52 | 78.45 86.58 |
| CVeDRL | **0.6B** | **72.14** | **78.72** | 78.96 **87.05** |
| **MBPP+** | | | | |
| Vanilla | – | 49.17 | 69.28 | 70.15 71.32 |
| MBR-E | – | 49.98 | 69.75 | 70.49 72.12 |
| CodeT | – | 59.17 | 69.88 | 69.93 73.28 |
| Llama3.1 | 70B | 65.24 | 71.77 | 75.87 75.01 |
| CodeRM | 8.0B | 66.63 | 72.53 | 75.98 75.18 |
| Qwen3 | 0.6B | 51.01 | 70.19 | 72.86 73.74 |
| Qwen3 | 32B | 65.43 | 72.61 | 75.71 75.04 |
| CVeDRL | **0.6B** | **66.79** | **73.60** | **76.21** **76.93** |
| **LiveCodeBench** | | | | |
| Vanilla | – | 11.74 | 25.19 | 20.48 34.90 |
| MBR-E | – | 12.12 | 25.18 | 20.51 34.79 |
| CodeT | – | 12.59 | 25.92 | 20.60 35.11 |
| Llama3.1 | 70B | 13.33 | **28.39** | 22.79 38.61 |
| CodeRM | 8.0B | 15.24 | 27.81 | 21.81 39.18 |
| Qwen3 | 0.6B | 12.52 | 25.47 | 21.91 37.18 |
| Qwen3 | 32B | 15.17 | 27.95 | 22.86 38.51 |
| CVeDRL | **0.6B** | **16.75** | 27.96 | **23.09** **39.31** |

Table 1: The result for code verification of CVe-DRL and other baselines over three code generation benchmarks. The top two performances for each dataset and policy model are marked in **bold** and underlined.

| LLM | Scale↓ | ER%↓ | FR%↓ | PR%↑ | BC%↑ | AN↓ |
| --- | --- | --- | --- | --- | --- | --- |
| **HumanEval+** | | | | | | |
| GPT-4o | – | 1.98 | 17.21 | 80.81 | 96.91 | 5.35 |
| GPT-3.5 | – | 3.14 | 26.32 | 70.54 | 96.73 | 4.13 |
| LLaMA3.1 | 8.0B | 10.88 | 37.19 | 51.93 | 94.60 | 3.97 |
| CodeRM | 8.0B | 2.44 | 64.73 | 32.83 | **96.97** | 7.15 |
| Qwen3 | 0.6B | 20.70 | 44.82 | 34.48 | 73.19 | 4.38 |
| Qwen3 | 32B | 9.43 | 25.31 | 65.26 | 89.53 | 8.17 |
| CVeDRL | **0.6B** | **1.27** | **12.79** | **85.94** | 97.53 | **2.41** |
| **MBPP+** | | | | | | |
| GPT-4o | – | 3.98 | 29.89 | 66.13 | 96.91 | 6.12 |
| GPT-3.5 | – | 5.14 | 40.15 | 54.71 | 96.65 | 5.97 |
| LLaMA3.1 | 8.0B | 15.79 | 47.53 | 36.68 | 95.93 | 4.13 |
| CodeRM | 8.0B | 2.44 | 52.86 | 44.70 | 97.11 | 7.88 |
| Qwen3 | 0.6B | 28.18 | 31.53 | 40.29 | 90.12 | 3.48 |
| Qwen3 | 32B | 11.42 | 31.39 | 57.19 | 92.44 | 7.47 |
| CVeDRL | **0.6B** | **0.53** | **15.79** | **83.68** | 97.37 | **3.13** |
| **LeetCode** | | | | | | |
| GPT-4o | – | **2.77** | 25.14 | 72.09 | 87.64 | 5.77 |
| GPT-3.5 | – | 3.53 | 37.28 | 59.19 | 86.53 | 5.63 |
| LLaMA3.1 | 8.0B | 12.47 | 54.77 | 32.76 | 81.49 | 3.88 |
| CodeRM | 8.0B | 3.70 | 58.16 | 38.14 | 85.37 | 6.43 |
| Qwen3 | 0.6B | 37.31 | 42.26 | 20.43 | 61.47 | 3.76 |
| Qwen3 | 32B | 23.18 | 27.63 | 49.19 | 78.62 | 7.34 |
| CVeDRL | **0.6B** | 3.49 | **20.98** | **75.53** | **91.61** | **2.84** |

Table 2: The direct testing results of CVe-DRL and other baselines with metrics including Error Rate(ER), Failure Rate(FR), Pass Rate(PR), Branch Coverage(BC) and Assertion Number(AN) across three datasets. Top two performances for each dataset and policy model are marked in **bold** and underlined.

and LEETCODE greengerong (2023). Following the methodology of CodeRM Ma et al. (2025), we select 168 function-style problems from LiveCodeBench created between January and September 2024. These tasks are known to challenge large models, making them suitable for stress-testing reward signals. To assess generation performance in broader contexts, we filter 2,360 LeetCode problems following VALTEST Taherkhani & Hemmati (2024) to exclude system-design questions, class-based interfaces, and interactive I/O. We yield 542 problems each defined by a single Python function signature with a return value. During training, we use the CODERM(Ma et al., 2025) dataset, which contains over 50,000 questions selected through a dedicated processing pipeline, and ensures that most of the chosen large model solutions are both appropriate and correct.

**Metrics.** We report five metrics for test quality evaluation:(1) Pass Rate (PR): the proportion of tests that execute without assertion errors. (2) Failure Rate (FR): the probability that a generated test runs successfully but the model's expected output is incorrect. (3) Error Rate (ER): the probability that the testing doesn't gives any coverage or failure report due to errors. (4) Branch Coverage (BC): the ratio of executed branches or lines covered by the tests. (5) Assertion Number (AN): the number of assertions within the unittest class. For the evaluation of validation, we adopt the pass-of-n metric under a unit-test majority voting framework, where the final solution is selected based on the number of test cases it successfully passes.

**Baseline Models.** In the evaluation of Validation-Coder Performance, we select four high-accuracy policy models including GPT-3.5, GPT-4o-mini Achiam et al. (2023), LLaMA-70B Dubey et al. (2024), and LLaMA-8B Dubey et al. (2024). For reward model inference, the baselines include CodeT Chen et al. (2023), MBR-E Shi et al. (2022a),and CodeRM-8B Ma et al. (2025). We also evaluate LLaMA-70B as a reward model to compare performance against larger unaligned models, and vanilla method with random selection of code solutions. For the evaluation of test quality, we employ GPT-4o Achiam et al. (2023), GPT-3.5 Brown et al. (2020), and LLaMA3-8B itellama3-

report as baseline models. To assess the gains provided by the CVeDRL training method, we report baseline model performance on both the direct evaluation and application evaluation tasks. We adopt Qwen3-0.6B-Base as the backbone model for training CVeDRL.

## 4.2 MAIN RESULTS

**Validation-Coder Performance.** Table 1 summarizes the outcomes of our code-verification experiments across three benchmark datasets. Following the majority-voting scheme, each model's candidate solutions are verified by running a fixed set of generated test cases. Compared to the pre-trained Qwen3-0.6B-base model, CVeDRL-0.6B significantly improves the best-of-$n$ performance on all benchmarks. Against the LLaMA-70B with much larger scale, CVeDRL-0.6B yields greater gains in nearly every experiment, demonstrating that its generated unit tests more effectively discriminate correct solutions. For example, when using CVeDRL-0.6B as the reward model to select among GPT-4o-mini outputs on the MBPP+ dataset, the accuracy rises from 71.32% to 76.93%, an improvement of 5.61%. CVeDRL attains first-tier performance across all evaluated policy models and datasets, with most of its metrics lying within a narrow margin of the second-best baselines. These findings indicate a potential performance ceiling inherent to the policy models and demonstrate that CVeDRL, as a code verifier, can effectively select superior solutions to further elevate policy model outputs. Although larger models (e.g., 1B or 3B parameters) could potentially offer improved accuracy, the chosen 0.6B scale strikes an optimal balance between verification performance and inference efficiency.

**Test Quality of CVeDRL.** Combining coverage reports with static code-analysis tools, we directly executed the generated tests and computed five evaluation metrics, the results of which are summarized in Table 2 for three benchmark datasets. By incorporating negative rewards for assertion errors and positive, difficulty-weighted coverage rewards during training, CVeDRL-0.6B maintains an exceptionally low failure rate and high line coverage, thereby robustly improving the quality of generated unit tests. On the MBPP+ dataset, CVeDRL-0.6B significantly outperforms every compared model, with the pass rate (PR) exceeding that of the next best model GPT-4o by 16%. Notably, CVeDRL-0.6B also produces significantly fewer assertions, indicating that our method minimizes redundant test cases while sustaining coverage levels and thus further reduces testing overhead. Moreover, Table 2 shows that CVeDRL-0.6B produces the highest-quality test suites on MBPP+, and correspondingly achieves the best reward-model performance from table 1 on that task compared to all baselines. This correlation between test-generation quality and reward-model effectiveness suggests a tight link between coverage gain and solution selection, which we analyze in detail in Appendix A.

## 4.3 SAMPLING EFFICIENCY OF CVEDRL

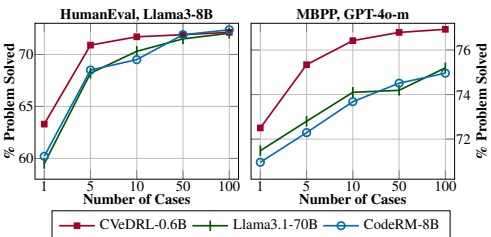

Figure 2: Performance of three unit-test generators at different test-case scales, with LLaMA3-8B on HumanEval+ and GPT-4o-mini on MBPP+ as policy model separately.

| LLM | Scale | AMU↓ (GB) | AL↓ (s/iter) | TT↑ (tok/kPar/s) | AEC↓ (W) |
|---|---|---|---|---|---|
| CodeRM | 8.0B | 43.07 | 1.7823 | 0.296 | 293.6 |
| Qwen3 | 4.0B | 41.70 | 4.1951 | 0.529 | 294.0 |
| Qwen3 | 0.6B | 40.97 | 1.3415 | 6.752 | 276.4 |
| **CVeDRL** | **0.6B** | **40.97** | **0.6622** | **7.083** | **246.2** |

Table 3: Inference efficiency comparison across models. For each model, we compare four key metrics: Average Memory Utilization (AMU), Average Latency (AL), Tokens Throughput (TT),and Average Energy Consumption (AEC). All results were obtained on the MBPP+ benchmark with 10 inference sampling per question.

Figure 2 illustrates the performance of three unit-test generators under varying test-suite scales across two policy models and datasets. Both plots demonstrate that increasing the number of sampled test cases generally improves reward-model efficacy, although the magnitude of improvement differs. Notably, CVeDRL-0.6B reaches its performance plateau with as few as 10 test cases in both

experimental settings, indicating minimal gains when sampling up to 100 cases under a majority-voting selection criterion. When using LLaMA3-8B as the policy model on HumanEval, all three reward models perform comparably at a scale of 100, but CVeDRL-0.6B significantly outperforms the other two baselines at a scale of 5. Similarly, with GPT-4o-mini as policy model on MBPP+, CVeDRL-0.6B consistently surpasses the other reward models at every scale. This is because the unit tests generated by CVeDRL are of high quality. As derived from the confidence bound, fewer majority-voting candidates are required, which accelerates the inference process. The detailed discussion is provided in the Appendix A.

## 4.4 INFERENCE EFFICIENCY OF CVEDRL

The results in Table 3 reveal clear scale-dependent trade-offs between resource utilization, latency, per-parameter throughput and energy efficiency. Relative to the SFT-trained CodeRM, CVeDRL-0.6B boosts token throughput from 0.296 to 7.083, which is an improvement of more than 20x. CVeDRL-0.6B further halves the latency from 0.66s/iter to 1.34s/iter compared with base model and marginally increases per-parameter throughput. CVeDRL-0.6B maintains competitive performance while significantly reducing computational requirements, thereby achieving a favorable trade-off between precision and latency, which aligns with the primary objective of rapidly verifying large amounts of generated code.

## 4.5 ABLATION STUDY

| Model | SYN | BDA | SDA | Mbpp+ | | | Humaneval+ | | |
|-------|-----|-----|-----|-------|-------|-----|-------|-------|-----|
| | | | | PR%↑ | BC%↑ | AN↓ | PR%↑ | BC%↑ | AN↓ |
| CodeRM | – | – | – | 44.70 | 97.11 | 7.88 | 32.83 | 96.97 | 7.15 |
| CVeDRL | × | × | × | 51.53 | 85.11 | 3.23 | 41.99 | 92.41 | 3.17 |
| | ✓ | × | × | 69.47 | 89.79 | 3.75 | 68.13 | 94.83 | 3.64 |
| | ✓ | × | ✓ | 79.96 | 92.14 | 3.47 | 84.14 | 94.55 | 3.88 |
| | ✓ | ✓ | × | 71.42 | 96.75 | 3.08 | 79.15 | 97.41 | 2.75 |
| | ✓ | ✓ | ✓ | **83.68** | **97.37** | **3.13** | **85.94** | **97.53** | **2.41** |

Table 4: Ablation study on the MBPP+ benchmark to disentangle the contributions of the exponent-shaped exploration for syntax reward("SYN"), branch-difficulty aware RL("BDA") and the static analysis reward for sample-difficulty aware RL("SDA").

We perform an ablation on CVeDRL-0.6B (Table 4) to isolate the effects of our exponent-shaped exploration for syntax reward, Branch-Difficulty aware RL and static analysis reward for Sample-Difficulty aware RL. For comparison of training methodologies, we also evaluate CodeRM-8B, a model trained on the same dataset but using supervised fine-tuning as training method. Static alone substantially improves pass rate by steering the agent toward appropriately difficult tasks, while exponential shaping significantly boosts code coverage by rewarding exploration of sparsely exercised branches. Significantly, combining both yields the highest overall performance, demonstrating their complementary benefits.

## 5 CONCLUSION

We introduce CVeDRL, a unified reinforcement learning framework that leverages static code complexity metrics and test results to guide effective unit-test generation and code solution verification. Through theoretical analyses, we establish explicit relationships between test-case pass rates, branch coverage, and overall verification performance. By integrating syntax and functionality considerations with branch-sample difficulty-aware mechanism, CVeDRL adeptly generates concise and comprehensive test suites capable of generalizing effectively across branches and complex code solutions. Experimental verification of diverse code generation benchmarks illustrates that CVeDRL achieves top-tier performance among various policy models. Overall, CVeDRL exemplifies the potential of reinforcement learning in advancing the efficiency and reliability of unit test generation, laying a strong foundation for future research directions in code verifiers.

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

# A  PASS–COVERAGE TRADE-OFF IN AUTOMATED CODE VERIFICATION

## A.1  SINGLE-TEST ASSUMPTIONS

In this subsection, we derive a concise relationship between the probability that a candidate program passes a test suite and the average branch coverage of the suite. We begin by introducing the notation for the single–test-case scenario, and then generalize our findings to the multi-test context via the mean coverage.

Let $q$ be the prior probability that a candidate program is *correct* and $p$ is the probability that a single *generated test* returns PASS on correct code, i.e., that the test's own assertion is valid. $r_{ij}$ is the probability that the test suite covers the $j$-th defective path (branch) of candidate $i$, where $n_i$ denotes the number of defective paths in candidate $i$. $c$ means branch coverage, typically approximated by $c = \frac{1}{mn} \sum_{i=1}^{m} \sum_{j=1}^{n_i} r_{ij}$; $0 \leq c \leq 1$. Then we come up with three assumptions as follows:

**Model Assumptions.**   We employ the following simplifying assumptions:

(a) **Correct code passes all exercised paths.** That is, if the test exercises any path of correct code, it will *pass* with probability $p$, and with probability $1 - p$ it fails due solely to a faulty assertion (a false negative).

(b) **Erroneous code has at least one faulty path.** If the test exercises one of these faulty paths *and* its assertion is valid, the error is revealed (the test fails as intended); otherwise the test yields a *false positive*.

(c) **Suite-level passing criterion.** A program is declared to *pass* the entire test suite and thus be emitted as a candidate if and only if it passes every individual test (the simplified pass_of_n rule).

**Single-Test Pass Probabilities.**   Under these assumptions, let $C$ be the event "program is correct". Then for a single test we have

$$P(\text{pass} \mid C) \,=\, p, \qquad P(\text{pass} \mid \bar{C}) \,=\, 1 - p\,r_i,$$

where

$$r_i = \sum_{j=1}^{n_i} r_{ij}$$

is the total probability that the test covers any one of the defective paths in candidate $i$. In practice, one often replaces $R_i$ by the average coverage $s$, yielding the approximation

$$P(\text{pass} \mid \bar{C}) \approx 1 - p\,c.$$

**Posterior Probability of Correctness.** Given that a candidate program has *passed* the test, we apply Bayes' theorem to update our belief:

$$P(C \mid \text{pass}) = \frac{P(\text{pass} \mid C)\,P(C)}{P(\text{pass} \mid C)\,P(C) + P(\text{pass} \mid \bar{C})\,P(\bar{C})}$$

$$= \frac{q\,p}{q\,p + (1-q)\big(1 - pr_i\big)}.$$

**Threshold for Improved Posterior.** We are particularly interested in the condition under which the posterior probability of correctness exceeds the prior:

$$P(C \mid \text{pass}) > q \quad \Longrightarrow \quad p > \frac{1}{1 + r_i}.$$

Substituting the mean coverage approximation $R_i \approx s$ yields the practical threshold

$$p > \frac{1}{1 + c}.$$

In other words, provided that the single-test reliability $p$ exceeds the reciprocal of one plus the average coverage, passing the test suite will *increase* our confidence that the program is indeed correct. This completes the derivation of the basic formula relating pass probability and coverage under our simplified single-test assumptions. Extensions to multiple independent tests follow by replacing $p$ and $r_i$ with their compounded quantities across the suite.

## A.2 MAJORITY VOTING FRAMEWORK

In this section, we extend our analysis to a *majority-voting* scheme. Let $N$ be the number of candidate programs (*best-of-$N$*) and $M$ be the number of independently generated test suites per candidate. $K$ denotes the number of I/O pairs in each suite, i.e. the number of assumptions across suites.

**Per-Suite Pass Probability.** When each suite contains $K$ I/O pairs, all assertions must pass for the suite to be considered successful. Hence

$$\begin{aligned} \alpha_c &= p^K && \text{for a correct program,} \\ \alpha_w &= (1 - pc)^K && \text{for an incorrect program.} \end{aligned}$$

where $1 - p\,c$ is the probability that either the defect path is not covered or the assertion fails to detect the error. We simplify our model in $K = 1$.

**Binomial Model for Passing Suites.** For any fixed candidate, the number of suites that pass follows a binomial distribution:

$$\mathbf{p} \sim \text{Binomial}\big(M, \alpha\big)$$

where

$$\alpha = \begin{cases} \alpha_c, & \text{if the program is correct,} \\ \alpha_w, & \text{if the program is incorrect.} \end{cases}$$

Under majority voting, we select the candidate with

$$s_{\text{opt}} = \arg\max_{i=1,\ldots,N} \mathbf{p}_i.$$

**Reliability via Concentration Bounds.** To ensure that the probability of selecting a wrong program is at most $\delta$, we require

$$P\big(\mathrm{s}_{\mathrm{opt}} \text{ is correct}\big) \; \geq \; 1 - \delta.$$

We now show how this follows from a Hoeffding-type concentration inequality applied to the difference in suite-pass counts between the true correct program and any incorrect one.

For each test index $t \in \{1, \ldots, M\}$ and each wrong candidate $j \in \{1, \ldots, WN\}$ where the wrong proportion of $N$ code solutions is $W$, define the indicator variables

$$\mathrm{x}_t = \mathbf{1}_{\text{test t passes on the correct solution}},$$

$$\mathrm{y}_{j,t} = \mathbf{1}_{\text{test t passes on wrong solution j}}.$$

By construction these are independent and identically distributed (i.i.d.), with

$$P(\mathrm{x}_t = 1) = \alpha_c, \qquad P(\mathrm{y}_{j,t} = 1) = \alpha_w,$$

where $\alpha_c$ and $\alpha_w$ are the per-suite pass probabilities for correct and incorrect code, respectively.

Let

$$\mathrm{g}_c = \sum_{t=1}^{M} \mathrm{x}_t, \qquad \mathrm{g}_j = \sum_{t=1}^{M} \mathrm{y}_{j,t},$$

$$\mathrm{d}_j = \mathrm{g}_c - \mathrm{g}_j = \sum_{t=1}^{M} \big(\mathrm{x}_t - \mathrm{y}_{j,t}\big) = \sum_{t=1}^{M} \mathrm{z}_{j,t}$$

where each summand $\mathrm{z}_{j,t} = \mathrm{x}_t - \mathrm{y}_{j,t}$ satisfies $\mathrm{z}_{j,t} \in [-1, 1]$. The expectation of the gap is

$$\mathbb{E}[\mathrm{d}_j] = \sum_{t=1}^{M} \mathbb{E}[\mathrm{z}_{j,t}] = M\big(\alpha_c - \alpha_w\big) = M\Delta.$$

By the one-sided Hoeffding bound for bounded independent random variables,

$$P(\mathrm{d}_j \leq 0) = P(\mathrm{d}_j - \mathbb{E}[\mathrm{d}_j] \leq -M\Delta)$$
$$\leq \exp\Big(-\tfrac{2(M\Delta)^2}{\sum_{t=1}^{M}(1-(-1))^2}\Big)$$
$$= \exp\Big(-\tfrac{M\Delta^2}{2}\Big) =: \beta.$$

Hence for each wrong candidate $j$, the probability that it ties or outperforms the correct program is at most $\beta$.

Then we calculate the union bound over all wrong candidates. Let $\mathcal{E}$ be the event that *any* of the $WN$ wrong programs achieves $\mathrm{s}_j \geq \mathrm{s}_c$, which indicates $\mathrm{s}_{opt}$ is incorrect. By the union bound,

$$P(\mathcal{E}) \; \leq \; WN\,\beta \; = \; WN \exp\Big(-\tfrac{M\Delta^2}{2}\Big).$$

Imposing the reliability target $P(\mathcal{E}) \leq \delta$ gives

$$WN \exp\Big(-\tfrac{M\Delta^2}{2}\Big) \leq \delta \quad \Longrightarrow \quad \Delta \; \geq \; \sqrt{\frac{2\ln\frac{WN}{\delta}}{M}}.$$

Recalling that $\Delta = \alpha_c - \alpha_w$ and $P\big(\mathrm{s}_{\mathrm{opt}} \text{ is correct}\big) \geq 1 - \delta$, we obtain the required "safety margin"

$$\alpha_c - \alpha_w \; \geq \; \sqrt{\frac{\ln\frac{WN}{\delta}}{2M}}$$

Operational bound for $K = 1$. In the single-case suite model ($K = 1$), we have $\alpha_c = p$ and $\alpha_w = 1 - p\,c$, hence

$$\Delta = p - (1 - p\,c) = p(1 + c) - 1.$$

Table 5: Estimated minimal number of independent test suites per candidate $M$ (rounded) for $N = 100$ with branch coverage $c$ of CVeDRL in Table 1, varying assertion reliability $p$. As an illustrative example, we adopt CVeDRL as the verifier and GPT-4o-m as the code generator, which is demonstrated in Table 2.

| Dataset | $p = 0.70$ | $p = 0.80$ | $p = 0.85$ |
|---|---|---|---|
| HumanEval+ ( $q = 0.8257 \rightarrow q' = 0.8705$, $c = 0.96$ ) | 71 | 30 | 21 |
| MBPP+ ( $q = 0.7132 \rightarrow q' = 0.7693$, $c = 0.97$ ) | 67 | 29 | 20 |

Then we have

$$p(1 + c) - 1 \ \geq \ \sqrt{\frac{\ln \frac{WN}{\delta}}{2M}}.$$

Substitution yields the practical requirement

$$p \ \geq \ \frac{1 + \sqrt{\frac{2}{M} \ln\left(\frac{WN}{\delta}\right)}}{1 + c}.$$

Finally, if we denote by $q' = 1 - \delta$ the overall probability of correct selection with majority voting framework and replace the fraction of wrong solutions $W$ by its expectation $E[W] = 1 - q$, the bound becomes

$$p \ \geq \ \frac{1 + \sqrt{\frac{2}{M} \ln\left(\frac{1-q}{1-q'}N\right)}}{1 + c} \tag{$\star$}$$

**Inference setting with majority-voting framework and confidence bound.** As an example, we show how the confidence bound quantifies the trade-off between test quality and the reduction in required test suites $M$ per candidate when selecting among multiple program proposals. The numerical entries in Table 5 ($q, q', c, p$) is obtained from the experimental tables, where the prior $q$ is the baseline ("Vanilla") correctness rate measured for the policy column, and the posterior target $q'$ is the improved correctness rate reported under CVeDRL (the "CVeDRL" row) from Table 1. The coverage parameter $c$ was taken from the verifier branch coverage statistics (BC) in Table 2 and used as a proxy for average branch coverage (hence $c \approx 0.96$ for HumanEval+ and $c \approx 0.97$ for MBPP+). The per-assertion reliability $p$ was treated as the variable in the sensitivity grid reported in Table 5. The operational bound ($\star$) indicates that

$$M \approx \frac{2 \ln\left(\frac{1-q}{1-q'}N\right)}{\left((1 + c)p - 1\right)^2}.$$

Improving the verifier (as CVeDRL does) reduces the minimal required number of independent suites $M$ through two channels: (i) raising the effective assertion reliability $p$ (signalled by higher PR and lower ER/FR) and (ii) increasing measured coverage $c$ (higher BC). Both actions enlarge the denominator $((1 + c)p - 1)^2$ and thus shrink $M$ (the dependence is approximately quadratic in the effective margin). However, the inequality also highlights a countervailing effect: specifying a higher target post-selection confidence $q'$ increases the logarithmic numerator $\ln\left(\frac{1-q}{1-q'}N\right)$, and therefore raises the required $M$. Fundamentally, we set $N = M = 100$ during inference time. In this case, the number of samples $M$ required for CVeDRL to attain the target correctness rate $q'$ is substantially less than 100. Notably, CVeDRL nearly reaches the desired performance at $M \approx 10$ as illustrated in Figure 2, which clearly surpasses the capability of other code verifiers. As shown in Table 5, CVeDRL achieves a pass rate close to 85%, with the required number of majority-voting candidates being around 20. This indicates that the target performance can be reached when $M \approx 20$, which is more than three times fewer samples than those required by a verifier with a unit test accuracy of $p = 0.70$ for both dataset.

## A.3 CONCLUSION

In summary, our analysis reveals that the interplay between test-assertion reliability $p$ and average branch coverage $s$ fundamentally determines the posterior confidence in a candidate program's correctness. A single test suffices to improve confidence only when $p > 1/(1 + c)$. With majority

voting framework across $M$ independent suites and $N$ candidates, the pass rate of unit-test generated by LLMs requires a stronger condition ($\star$). These bounds quantify explicit trade-offs. More test suites or higher coverage can compensate for imperfect assertions, whereas larger candidate pools or stricter error tolerances demand more reliable tests. This provides principled guidelines for designing automated testing pipelines that balance resource expenditure against desired selection reliability. Specifically, the analysis is utilized for choosing the crucial hyperparameter $M$ which is relative to acceleration.

## B  Additional Discussions

### B.1  Impact of Ancillary LLM Components

To investigate how the choice of base model influences the effectiveness of the CVeDRL training method, we include Qwen2.5-Coder-0.5B as a benchmark against Qwen3. Additionally, we perform an ablation study by disabling Qwen3's chain-of-thought capability using the `/no_think` tag. Finally, motivated by prior findings that large LLMs struggle with static code analysis, we augment the prompts with conditional branch information to assess whether this auxiliary context yields further performance gains.

The ablation study in Table 6 indicates that prompting primarily guides the model toward correct solutions with fewer attempts, while chain-of-thought (CoT) encourages broader exploration at the expense of efficiency. In the smaller Qwen2.5 model, prompts reduce the average number of trials but slightly lower overall pass rate, whereas disabling prompts yields higher pass rate with more attempts. Although Qwen2.5 shows reduced pass rate and coverage relative to Qwen3, it requires fewer generated assertions and nonetheless significantly outperforms GPT-4o, highlighting the superiority of the CVeDRL training methodology. The ablation results show that introducing chain-of-thought (CoT) consistently harms overall performance, lowering both pass rate and coverage. Moreover, adding conditional prompts has no appreciable effect on branch coverage but nonetheless reduces the model's success rate. These findings indicate that while CoT and extra prompt constraints aim to guide reasoning, they in fact impede efficiency without delivering coverage benefits.

Table 6: Ablation of Prompt and CoT on Validation-Coder Performance

| Base | Prompt | CoT | PR% | BC% | AN |
|------|--------|-----|------|------|------|
| Qwen2.5 | ✓ | \ | 71.68 | 96.79 | **2.09** |
| Qwen2.5 | × | \ | 75.43 | 96.86 | 2.53 |
| Qwen3 | × | × | **82.01** | 97.14 | 2.91 |
| Qwen3 | × | ✓ | 80.37 | 96.15 | 3.47 |
| Qwen3 | ✓ | × | 79.98 | **97.21** | 3.13 |
| Qwen3 | ✓ | ✓ | 79.71 | 96.54 | 3.64 |
| GPT-4o | \ | \ | 66.13 | 96.91 | 6.12 |

### B.2  Limitation

**Partial code support.**  In real-world code generation tasks such as code completion, the generated code is often partial and thus cannot be directly validated by unit tests. Although the sample-branch syntax/functionality reward we propose can be adapted to handle partial programs and library-level (test-suite) training once a suitable development environment and dependencies are provisioned, this capability has not yet been integrated into our pipeline. Future work will close this gap by incorporating partial-code validation into the pipeline and extending the approach to support additional programming languages and broader library ecosystems.

**Unit test adaptability.**  While our experiments show that the current unit tests generated by CVe-DRL are effective at filtering incorrect code, the verifier itself cannot distinguish the intrinsic cor-

rectness of the code solution. Future work should incorporate code mutation or auxiliary principles to alleviate this limitation.

## C  TRAINING CONFIGURATION

**Framework and Algorithm.**    We fine-tune the Qwen3-0.6B and Qwen2.5-Coder-0.5B checkpoint with the `verl` RL library using *Group Relative Policy Optimization* (GRPO). `verl` is an open-source reinforcement-learning framework designed for post-training fine-tuning of large language models, providing a hybrid single- and multi-controller programming model for scalable PPO and GRPO workflows. It features modular APIs that decouple computation and data dependencies and integrates seamlessly with PyTorch FSDP, Megatron-LM, vLLM, and other LLM infrastructures for efficient, production-ready deployments. GRPO eliminates the separate value network and updates the policy by comparing each sampled trajectory to the within-group reward baseline, thereby reducing both memory footprint and wall-clock cost.

**Dataset.**    All prompts are taken from the publicly-available CodeRM-UnitTest corpus, which provides 17,600 training items and 59,600 held-out test items. The CodeRM-UnitTest dataset is a curated collection of over 77 000 synthetic Python unit tests, derived from CodeFeedback-Filtered-Instruction and TACO, and provided in Parquet format for training test-guided code-reward models. It serves as the primary training and evaluation corpus for lightweight unit-test generator models like CodeRM-8B, enabling rigorous performance benchmarks under real execution feedback. Because every roll-out requires real execution of unit tests, we randomly subsample 3,000 test cases for validation to keep evaluation under two hours per checkpoint. Experiments ran on $2 \times$ NVIDIA A100 40 GB GPUs (FP16) with total training time of roughly 48 hours. Table 7 summarises the hyper-parameters that most influence optimization and compute. All other settings follow the official `verl` GRPO recipe.

| Parameter | Value |
|---|---|
| Learning rate | $1 \times 10^{-6}$ |
| Global prompt batch size | 32 |
| Rollouts per prompt | 2 |
| Max prompt / response length | 6 150 / 2 048 tokens |
| Mini-batch / micro-batch size | 16 / 8 |
| KL loss coefficient | 0.001 (low-variance) |
| Entropy coefficient | 0 (disabled) |
| GRPO clip ratio | 0.2 (default) |
| Total epochs | 1 000 |
| Gradient checkpointing | enabled |
| FSDP offload | disabled |

Table 7: Salient hyper-parameters used in GRPO fine-tuning.

## D  EXPERIMENTAL DETAILS

**Policy models.**    We evaluate four instruction-tuned policy models of different capacity and provider type: *Llama3-8B-Instruct*, *Llama3-70B-Instruct*, *GPT-3.5-turbo*, and *GPT-4o-mini*. Each model produces at most 100 candidate code solutions per prompt. Decoding and verification are run on $4 \times$ NVIDIA A100-40 GB GPUs.

**Baselines.**    We consider three verification-oriented baselines that exploit unit-test feedback to discriminate among candidate programs.

- Vanilla : the top-1 sample of the policy model without any reranking.
- MBR-E : minimum-Bayes-risk decoding that ranks candidates by the empirical risk computed from the execution outcomes of LLM-generated test cases (we use the "hard-loss" variant).

- CodeT : dual-execution agreement that measures both (i) consistency between candidate programs and their generated tests and (ii) cross-candidate concordance.

Besides the three test-driven methods discussed above, we measure performance against four capacity-oriented baselines: an 8B reward model CodeRM-8B for score-based reranking, a strong 70B code LLM Llama3-70B-Instruct, the supervised-fine-tuned backbone Qwen3-0.6B (Base), and our model CVeDRL-0.6B.

- CodeRM-8B : an 8B-parameter reward model that assigns a scalar quality score to each candidate; we select the highest-scoring solution.
- Llama3.1-70B : a strong open-source coder whose single best sample is taken as a standalone baseline.
- Qwen3-0.6B (Base) : the supervised-fine-tuned version of our backbone model, without reinforcement learning.
- CVeDRL-0.6B : our method, trained with GRPO on weighted data (see Appendix C).

**Datasets.** To obtain a representative view of unit-test generation, we aggregate four publicly available benchmarks: HumanEval+ and MBPP+, LiveCodeBench from Jan to Sep 2024, and the algorithm-oriented subset of LeetCode. Together, they cover both synthetic interview-style problems and organically authored, in-the-wild code, furnishing a diverse test bed for large-language-model (LLM) evaluation.

- LivecodeBench : Following the CodeRM protocol, we retain the 168 function-style tasks released between January – September 2024, because these newer problems have empirically proved the most challenging for current LLMs.
- LeetCode : Starting from 2,360 publicly accessible problems, we apply the VALTEST filtering rules to discard system-design, class-interface, and interactive I/O questions. The remaining 542 tasks each expose one Python function signature with a deterministic return value, enabling uniform test-harness construction.

For every benchmark we normalise signatures, strip extraneous boilerplate, and compile tests into a unified execution harness so that success rate, failure rate, branch coverage, and Pass@N can be measured consistently across datasets.

**Experimental Rationale.** HUMANEVAL+ and MBPP+ are retained in both the CVeDRL Test Quality and Validation-Coder Performance studies in main result because their moderate task counts, authoritative reference solutions, and fine-grained coverage tooling permit reliable intrinsic scoring while keeping the compute requirements of iterative Pass@N evaluation tractable. The filtered LEETCODE subset is confined to the Test Quality analysis. Its larger problem set and costly edge-case generators provide valuable stress-testing for success, error, and coverage metrics, but render exhaustive validation-coder search computationally infeasible. Conversely, LIVECODEBENCH appears only in the Validation-Coder study. Its developer-written tests supply a strong external oracle for synthesis evaluation, yet their heterogeneity prevents fair aggregation with coverage-based quality metrics.

Due to the constraints of the double-blind review policy, the model weights are not publicly released at this stage.

1026
1027
1028
1029
1030
1031
1032
1033
1034
1035
1036
1037
1038
1039
1040
1041
1042
1043
1044
1045
1046
1047
1048
1049
1050
1051
1052
1053
1054
1055
1056
1057
1058
1059
1060
1061
1062
1063
1064
1065
1066
1067
1068
1069
1070
1071
1072
1073
1074
1075
1076
1077
1078
1079

**[Original Prompt](Input)**

Below is a question and it's corresponding code answer. Please write test cases to check the correctness of the code answer. You need to use the unittest library in Python and create a test class for testing.

### question
Subodhians are taught to add multi-digit numbers from right-to-left one digit at a time.Many find the \"carry\" operation - in which 1 is carried from one digit position to be added to the next - to be a significant challenge. Your job is to count the number of carry operations for each of addition problem so that educators may assess their difficulty.\n\nFor the input first line contains n number of records which is less then 1000. And then each line contains two numbers with spacing between them.

### code solution
```python
def count_carry_operations(a: str, b: str) -> int:
    # Ensure both strings are of equal length by padding with leading zeros
    max_len = max(len(a), len(b))
    a = a.zfill(max_len)
    b = b.zfill(max_len)

    carry_count = 0
    carry = 0

    # Iterate from the last digit to the first
    for i in range(max_len - 1, -1, -1):
        digit_sum = int(a[i]) + int(b[i]) + carry
        if digit_sum >= 10:
            carry = 1
            carry_count += 1
        else:
            carry = 0

    return carry_count
```

**[Unit Test Output](Model Response)**

Here is the test case for the `count_carry_operations` function:

```python
import unittest

class TestCountCarryOperations(unittest.TestCase):
    def test_example1(self):
        self.assertEqual(count_carry_operations("123", "456"), 0)

    def test_example2(self):
        self.assertEqual(count_carry_operations("555", "555"), 3)

    def test_example3(self):
        self.assertEqual(count_carry_operations("123", "594"), 1)

if __name__ == '__main__':
    unittest.main()
```

This test case checks the function with the provided sample inputs and outputs.<|endoftext|>

**Test Coverage**

```
-------------------------------------------------------------------
...                                          [100%]
________ coverage: platform linux, python 3.10.16-final-0 ______

Name    Stmts  Miss  Cover  Missing
------------------------------------
src.py    13     0    100%
------------------------------------
TOTAL     13     0    100%
3 passed in 0.02s

{'coverage': '100%', 'uncovered_lines': {'src.py': ''}, 'failure': False,
'error': False, 'assertion_info': {'assertEqual': 3}}
{'coverage': 1.0, 'assertion_info': {'assertEqual': 3}}

-------------------------------------------------------------------
 Answer score: 1.0
 Format score: 1.0
 Total score: 2.0
-------------------------------------------------------------------
```

Figure 3: A case of the training pipeline for CVeDRL.