# OpenReview forum: "CVeDRL: An Efficient Code Verifier via Difficulty-aware Reinforcement Learning"
_ICLR.cc/2026/Conference — ICLR 2026 Conference Withdrawn Submission_

### Official Review · Reviewer_Xpb5 · 2025-10-27

**Soundness:** 3
**Presentation:** 3
**Contribution:** 2
**Rating:** 6
**Confidence:** 4

**Summary:**

The paper proposes CVeDRL, a 0.6B-parameter code verifier trained with reinforcement learning (GRPO) to generate higher-quality unit tests for sample-and-verify code generation. Key ideas include (i) a two-part reward (syntax validity + functionality / coverage), (ii) exponential reward shaping to emphasize rare/edge branches, and (iii) difficulty-aware training at both branch and sample levels using static analysis metrics (Halstead complexity, Maintainability Index). On HumanEval+, MBPP+, and other benchmarks, the learned verifier purportedly attains better verification power with fewer test cases and lower latency than SFT baselines (e.g., CodeRM-8B).

**Strengths:**

1.Practical problem: improving verifiers directly benefits real code-gen systems via accuracy and cost/latency reductions.
2. Well-motivated design: syntax + functionality rewards are sensible; exponential shaping for rare branches is intuitive; difficulty-aware weighting is simple yet effective.
3. Clear deployment story: the verifier is RL-trained offline and then used at inference as a drop-in judge; the paper cleanly separates training vs usage.
4. Efficiency: empirical results suggest comparable or better selection accuracy with notably fewer tests and lower inference latency than stronger SFT verifiers.

**Weaknesses:**

1. External validity of difficulty metrics: Halstead and maintainability indices may not generalize across languages or code styles; broader language coverage is limited.
2. Theory scope: the majority-vote reliability bound is helpful but presented at a high level; assumptions (e.g., independence of test outcomes across groups) are not tested.
3. Ablations depth: more analysis on shaping schedules, learned difficulty predictors vs static metrics, cross-language generalization, and sensitivity to generator strength would be valuable.
4.The main baseline,CodeRM is based on Llama-3.1 and CVeDRL is based on Qwen3, which is more competitive. They need to use the same base model to ensure fairness.

**Questions:**

1. How sensitive are gains to the exponential shaping schedule? Have you tried learning the shaping parameters?
2. Can the difficulty component be replaced by a learned predictor trained from outcomes?
3. Any cross-language experiments or tasks with heavy I/O, randomness, or concurrency?

---

### Official Review · Reviewer_nBrP · 2025-10-30

**Soundness:** 3
**Presentation:** 2
**Contribution:** 2
**Rating:** 4
**Confidence:** 4

**Summary:**

The paper proposes a reinforcement learning-based framework to train compact language models for code verification through unit-test generation. Instead of relying on supervised fine-tuning, CVeDRL uses Group Reward Policy Optimization (GRPO) with a composite reward that balances syntactic correctness and functional execution results. It further introduces two more reward components: branch-difficulty-aware rewards and sample-difficulty-aware rewards. The paper evaluates their approach on four benchmarks with four instruction-tuned policy models and shows improvement in pass rates and branch coverage with faster inference efficiency.

**Strengths:**

- Novel reward design: Combines syntax–functionality rewards with branch-difficulty-aware and sample-difficulty-aware reinforcement learning, a formulation not seen in prior RL-for-testing literature.

- Theoretical grounding: Derives a quantitative bound linking test-case reliability, branch coverage, and verification confidence — providing rare analytical rigor for code-verification RL work.

- Empirical gains: Achieves superior pass rate, branch coverage, and efficiency (20× faster inference) compared with SFT-based baselines and even larger models like GPT-4o-mini or LLaMA-70B.

**Weaknesses:**

- Limited novelty in RL backbone: Uses GRPO (a known variant of DPO) with mostly standard RL fine-tuning procedures; the conceptual innovation mainly lies in reward shaping rather than algorithmic foundations.
- Lack of evaluation in various model architectures: The paper claims to have good performance on “0.6B” scale models, but only evaluated their approach on Qwen3-0.6B-base model. Testing on more model architectures with a similar number of parameters will help justify the claim.
- Lack of evaluation on different model sizes: How does your RL framework perform on models with different sizes?
Values not justified: In section 3.2 Functionality Reward section, the choices for “-2.0” and “-1.5” are not well justified. Why these two numbers?
- Missing metrics in ablation study: In section 4.5 ablation study, only three of five metrics are reported. Where are the ablation results for metrics “FR” and “ER”?
- Lack of on-par SFT baselines: You are claiming a better performance and efficiency compared to SFT approach. However, no baseline for SFT approach on the same Qwen3-0.6B-base model is provided as a comparison.
- Writing format: The paper does not include numbers for equations.

**Questions:**

- See the Weaknesses Section
- How long does RL training take?

---

### Official Review · Reviewer_WT5y · 2025-10-31

**Soundness:** 2
**Presentation:** 3
**Contribution:** 2
**Rating:** 4
**Confidence:** 3

**Summary:**

This paper introduces CVeDRL, an efficient code verifier trained using reinforcement learning to improve LLM code generation through better unit test generation and code verification. Existing SFT methods for code verification suffer from scarcity of high-quality training data, high error and failure rates in generated unit tests, and severe inference delays due to repeated sampling. The authors derive a mathematical relationship between test case pass rates, branch coverage, and code verifier performance, establishing a confidence bound that guides design decisions. They implements the CVeDRL Framework, a reinforcement learning approach using GRPO with two key reward components: 1. Syntax reward: proper formatting and AST structure. 2. Functionality reward: based on test execution outcomes (errors, failures, passes) weighted by code coverage. For the reward function, it uses exponential reward shaping to encourage testing of rare/boundary conditions rather than just easy-to-cover paths. Besides, it also incorporates static code analysis metrics (Halstead Complexity and Maintainability Index) to weight rewards based on code complexity.
CVeDRL-0.6B (only 600M parameters) achieves competitive or superior performance compared to GPT-4o-mini and GPT-3.5 with 17.55% higher test pass rate than GPT-4o-mini on MBPP+. 20x improvement in inference efficiency compared to SFT-based model CodeRM.

**Strengths:**

1. The combination of software engineering quality metrics with the GRPO rewards is creative.
2. The motivation of this paper is convincing. Verification generated by LLMs is an important part for verifying the code generated by LLMs.
3. The experiments of this paper is thorough, which cross different models and benchmarks.

**Weaknesses:**

1. The verification model has a narrow scope on the test cases it can generate. I'm not sure how would it generalize to larger problem in repo-level coding.
2. The training relies on code solutions that are appropriate and correct, which however is not very accessible for new domains of coding.

**Questions:**

1. You train by executing generated tests against LLM-generated code solutions from the CodeRM dataset, which you state contains solutions where "most are appropriate and correct." Can you quantify what percentage of CodeRM training solutions actually pass the canonical test suites on HumanEval+ and MBPP+ (where ground truth exists)?
2. How do you generalize your methods to other specific domain like scientific computing?

---

### Official Review · Reviewer_bqYs · 2025-11-01

**Soundness:** 3
**Presentation:** 3
**Contribution:** 2
**Rating:** 4
**Confidence:** 3

**Summary:**

The paper introduces CVeDRL, a reinforcement learning-based framework for training efficient code verifiers that generate unit tests to validate LLM-generated code. The paper proposes a 0.6B parameter model that combines syntax and functionality rewards with novel branch-difficulty and sample-difficulty aware mechanisms, incorporating static analysis metrics (Halstead Complexity and Maintainability Index) for improved reward shaping.

Overall, the paper makes incremental rather than breakthrough contributions. While static metrics are efficient, the paper doesn’t explain why these specific metrics matter or prove that they are necessary (better than other reward shaping mechanisms). The evaluation is also quite narrow, restricted to saturated Python benchmarks. The paper needs to better explain why these metrics improve learning, empirically and/or theoretically, and validate that the “better” code actually runs better.

**Strengths:**

- CVeDR 0.6B achieves good inference efficiency improvements (>20x throughput) compared to SFT-based models like CodeRM 8B, while maintaining competitive performance.
- The paper records impressive performance across benchmarks, such as 83.68% pass rate on MBPP+ despite its size.
- The exponential reward shaping for branch coverage and integration of static complexity metrics represents a creative approach to addressing boundary conditions.

**Weaknesses:**

- While the paper describes how HC and MI are integrated, it lacks rigorous justification for why these specific metrics improve upon existing dynamic reward approaches (no comparative studies, see point below). The geometric mean combination appears ad-hoc without theoretical grounding.
- The paper lacks comparisons with other RL-based verifiers of similar scale using dynamic rewards. This omission makes it difficult to isolate the contribution of the static difficulty metrics.
- The paper focuses on verification performance but doesn't validate whether the selected code solutions actually exhibit better functional quality or runtime performance beyond passing tests.
- All experiments are restricted to Python on saturated benchmarks, limiting claims about language agnostic effectiveness, especially on “real-world” repositories

**Questions:**

- Can you provide theoretical or empirical evidence that HC and MI capture something fundamental that execution-based rewards miss? Have you tried compared against other static metrics such as cyclomatic complexity alone, code churn, etc?

- You show CVeDRL selects different solutions than other verifiers, but are these solutions actually better in terms of runtime, memory usage, or readability? Do you have results via human (expert) evaluation?

- In Appendix B.2, the paper claims that it "cannot distinguish the intrinsic correctness of the code solution," which seems contradictory to its core purpose as a verifier. Can you clarify what you mean by this

---

### Note · Authors · 2025-11-13

I have read and agree with the venue's withdrawal policy on behalf of myself and my co-authors.